# On-chip Cherenkov radiation tuning in 3.2-14 THz

Tianchang Li[1,2], Fang Liu ®[1,2] ✉, Yudi Chen[1], Xiaotong Xiong[1], Kaiyu Cui ®[1], Xue Feng ®[1], Wei Zhang ®[1] & Yidong Huang ®[1] ✉

Cherenkov radiation has attracted much attention for promoting the free electron radiation sources. Using hyperbolic metamaterial, Cherenkov radiation can be excited using low-energy electrons and thus on-chip free electron radiation source has been realized. However, direct experimental observations of on-chip free-electron-based Cherenkov radiation have been limited to the visible region, and the tunability has not been thoroughly explored. In this work, by constructing a hyperbolic metamaterial with graphene and hexagonal boron nitride, on-chip, free-electron-based terahertz Cherenkov radiation is observed and the frequency tunable range spans 3.2 to 14 THz in a hundred-micron-scale dimension. Compared with other free-electron terahertz sources, the chip size is over three orders of magnitude smaller, and the tunable range is one of the widest. This work extends the on-chip free-electron-based Cherenkov radiation into terahertz domain, highlighting its tunability and paves the way for further advancements in free electron radiation source.

Cherenkov radiation (CR) is a kind of electromagnetic radiation generated by charged particles with velocity exceeding light velocity in the medium[1,2], which promotes the development of particle physics and radiation sources[3–5]. Since its discovery, efforts have been made to reduce the velocity threshold of charged particles for generating CR[6–8]. After applying hyperbolic metamaterial (HMM)[9–12], a kind of artificial material with signs of permittivity opposite in different directions, the electron energy for exciting CR is reduced in experiment, and an on-chip CR emitter is realized spanning the visible-infrared frequency range[5]. Although subsequent research on free-electron-based CR in HMM at other frequency ranges has been conducted theoretically and indirect detection has been acieved[13–15], until now, on-chip free-electron-based CR outside of the visible-infrared frequency range has still not been directly observed in experiment, and the frequency tunability of CR in HMM has not been thoroughly explored.

Terahertz (THz) waves have broad application prospects, and free-electron THz sources are an important class of THz radiation sources[16–20]. For the free-electron THz sources, such as accelerator-based free-electron lasers (FELs)[21,22] klystrons[23], orotrons[24], traveling wave tubes[25], backward-wave oscillators[26], and gyrotrons[27], on-chip integration cannot be achieved limited by the high kinetic energy of electrons, the size of the electron emitter, and magnetic components. Additionally, regarding frequency tuning, even though FELs can achieve a wide tunable frequency range, the tuning method of manipulating the electron beam by adjusting the chicane requires large-scale equipment[21,22]. Moreover, other kinds of free-electron THz sources, especially the compact sources, have much weaker frequency tunability since the fixed slow-wave structure supporting single or very few modes cannot be adjusted[25,26]. Therefore, a small-sized or on-chip integrated wide-range tunable free-electron THz radiation source has still not been reported.

Here, we report the direct observation of the on-chip free-electron-based CR in the THz frequency region and the on-chip integrated free-electron THz radiation source with superwide frequency tunability. By constructing a multilayer HMM with graphene and hexagonal boron nitride (hBN), CR is generated by low-energy electrons (1.4–2.6 keV), and the maximum frequency tunable range of 5 THz is achieved by only adjusting the voltage applied on the chip. Furthermore, by adopting the array structure, the frequency tunable range of the on-chip free-electron THz source spans 3.2–14 THz, which is at the same level as that of accelerator-based THz source[22] and much wider than that of most other free-electron THz sources[23–27]. The core area of

[1]Department of Electronic Engineering, Tsinghua University, Beijing, China. [2]These authors contributed equally: Tianchang Li, Fang Liu. ✉e-mail: liu_fang@tsinghua.edu.cn; yidonghuang@tsinghua.edu.cn

the chip ($\sim100 \times 100\ \mu m$) is more than three orders of magnitude smaller than that of all other kinds of free-electron THz sources, with an output power of 400 nW from the chip and corresponding power density as high as $10^5\ W\cdot m^{-2}$.

## Results

### Design and fabrication

The proposed on-chip free-electron THz source is sketched in Fig. 1a. The HMM in the THz frequency region is constructed by alternating graphene and hBN layers (see Supplementary Information Section S1). On the two sides of the source, the planar molybdenum (Mo) cathode with zigzag ends emits a belt-like free-electron beam in the $y$-direction, with a width of $\sim50\ \mu m$ and current of 10–70 μA when the voltage lies in 1.4–2.6 kV (see Supplementary Information Section S3). Traveling on top of HMM, the low-energy free-electron beam can generate CR in the THz frequency region. To control the gap and avoid collision between the electron beam and the HMM, the Mo electrodes are lifted by a layer of SiO$_2$ with a thickness of 350 nm. The free-electron-excited THz CR in the HMM is then extracted into free space by a metallic grating.

The generation of on-chip CR in graphene-hBN HMM by low-energy electrons can be understood by the 3D sketch of the wavevector matching. Figure 1b illustrates the dispersion relation of the evanescent field surrounding the free-electron beam (black line; $|\mathbf{k_z}| = \omega/u_0$, where $\omega$ is the angular frequency and $u_0$ is the electron velocity) and CR mode in the HMM (red surface; $k_x\text{-}k_z\text{-}\omega$), respectively. For a certain frequency $\omega_0$, CR with wavevector $\mathbf{k_{CR}}$ (blue arrow) can be excited for the longitudinal wavevector $\mathbf{k_z}$ matching with that of the evanescent field surrounding the free-electron beam. Since the CR modes in the HMM satisfy $|\mathbf{k_z}|>|\mathbf{k_{z\text{-}min}}| = \omega\sqrt{\varepsilon_x}/c$ (the end of $\mathbf{k_{z\text{-}min}}$ is represented as the red hollow dot in Fig. 1b), the velocity of free electrons $u_0$ of exciting CR in HMM should satisfy the condition $u_0 < c/\sqrt{\varepsilon_x}$ [5], where $c$ is the light velocity in vacuum and $\varepsilon_x$ is the permittivity component in the $x$-direction (for more detailed analysis, see Supplementary Information Section S2). Meanwhile, while considering the nonlocality of graphene[28,29] and the homogenization condition of HMM[5], there exists a lower electron energy bound for exciting CR in the multilayer HMM. After wavevector matching, CR in the HMM excited by the electrons has a wavevector $|\mathbf{k_z}| = \omega/u_0$ in the $z$-direction and then extracted into free space along $-x$ by the metallic grating according to the relation $\omega/u_0 = 2\pi n/p$, where $p$ is the grating period and the integer $n$ denotes the diffraction order. Additionally, gold is adopted as the material of the grating because it shows metallic

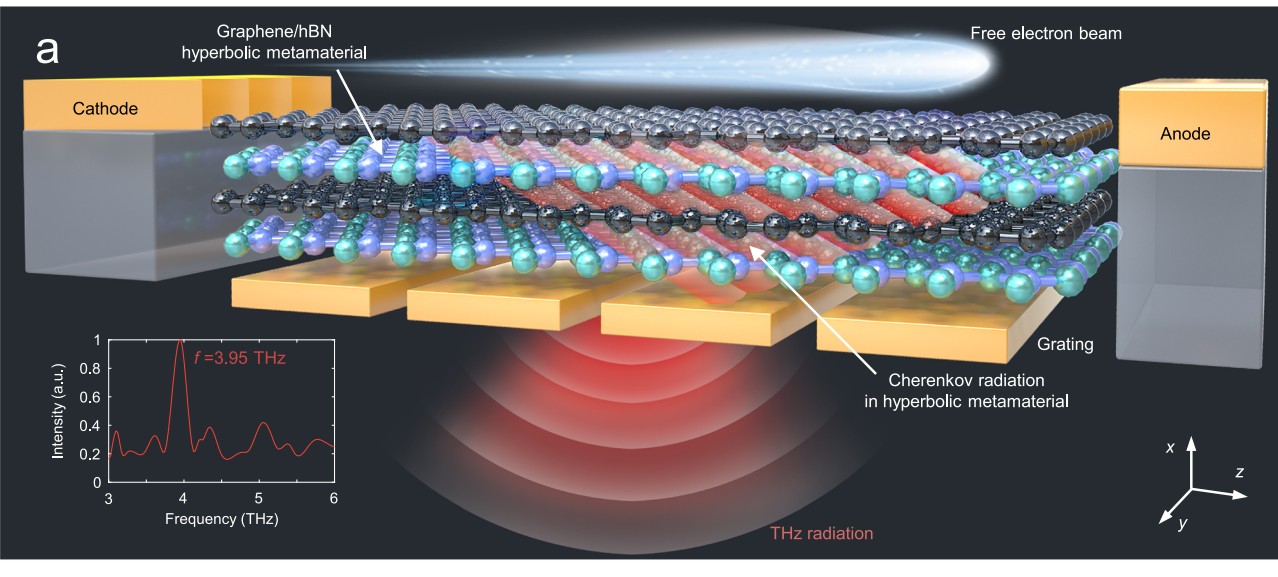

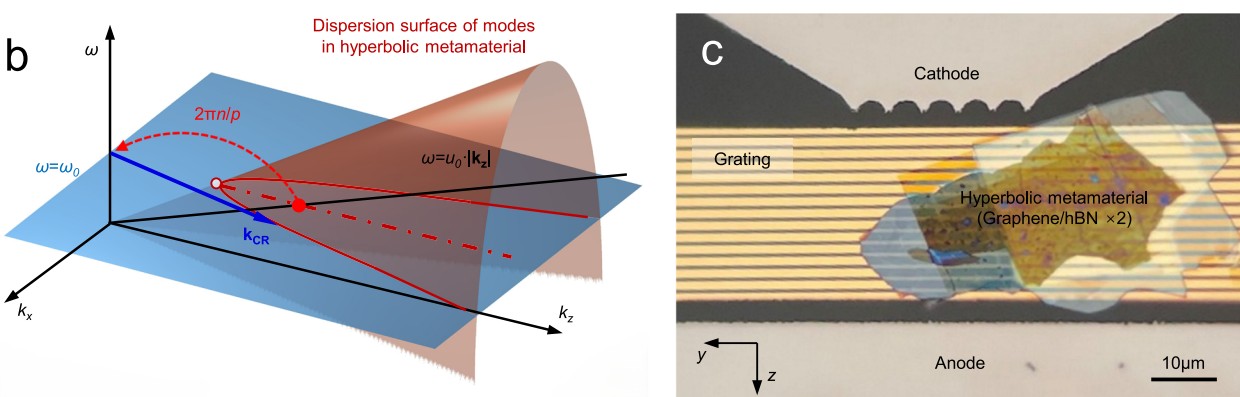

**Fig. 1 | On-chip integrated Cherenkov radiation emitter in the THz frequency region. a** Schematic diagram of the THz radiation chip. An HMM (alternating graphene/hBN 4-6 layer structure with a thickness of 100–200 nm and an area of $30 \times 30\ \mu m$) is adopted to generate broadband THz CR when excited by a free-electron beam. The field-emission Mo cathode and anode, with a thickness of ~350 nm and a distance of 50 μm, serve as the planar free-electron emitter and collector. A metallic grating with a period of $p = 2$–6 μm is applied to extract the CR in the HMM into free space along the $-x$-direction, with a frequency related to the electron energy and grating period. **b** 3D sketch of the wavevector matching of the evanescent field surrounding the free-electron beam (black line, $\omega = u_0\cdot|\mathbf{k_z}|$) to the CR mode in the HMM (blue arrow) and then to the free space wave. The red and blue surfaces represent the dispersion surface ($k_x\text{-}k_z\text{-}\omega$) of the modes in the HMM and the isofrequency plane (namely, $\omega = \omega_0$), respectively. **c** Optical microscope image of the radiation chip with cathode-anode and a four-layer graphene/hBN HMM transferred onto a gold grating.

properties in the THz region, and thus the grating shows a high contrast property, which can achieve large in-plane scattering, especially with a limited number of grating periods. After the grating extraction, only a relatively narrow band THz wave propagates along the $x$-direction orthogonal to the electron trajectory, and its frequency is determined by the electron velocity $u_0$ (kinetic energy $E$) and the grating period $p$.

Figure 1c shows optical microscopy images of the radiation chip with the graphene-hBN HMM. The cathode and anode are fabricated with a separation distance of 50 µm in the $z$-direction, between which the metallic grating and the ~30 × 30 µm HMM are located. The HMM is realized by alternatively stacking four to six layers of graphene (thickness of 1–5 nm) and hBN (thickness of 40–45 nm), with a total thickness of 100–200 nm. After thermal annealing, the HMM shows great flatness, as shown by the optical microscope image in Fig. 1c. (See "Methods" for more details of the fabrication.) With a homemade vacuum experimental system (for more details about the system, see "Methods" subsection "Measurement setup"), THz CR is detected with maximum power up to 100 nW (corresponding to ~400 nW emitted power, see Supplementary Information Section S4), and the voltage lying in 1.4–2.6 kV satisfies the on-chip CR condition.

## Measurement

Based on the radiation frequency relation $f = u_0/p$ (simplified from $\omega/u_0 = 2\pi n/p$ when $n = 1$), the range of $f$ can be obtained from the experimentally achievable ranges of $E$ and $p$. Figure 2a illustrates the radiation spectrum of the chip. By changing the electron energy $E$ in the range of 1.4–2.6 keV with fixed grating period $p$, the electrically tunable range is about 1.2–5 THz. The maximum tunable range of 5 THz is realized when $p = 2$ µm. Furthermore, by adopting the array structure with grating period $p$ of the units changing from 2 to 6 µm, the radiation frequency can cover a superwide range of 3.2–14 THz, and the central frequencies of the peaks are given at the top of Fig. 2a. The sample numbers are marked at the bottom of the figure, and the corresponding parameters $p$ and $E$ can be found in Supplementary Information Section S5. Moreover, a radiation power of up to 80–100 nW is detected for each radiation peak in Fig. 2a. Considering the collection and transmission loss along the optical path, the output power from the chip is estimated to be ~400 nW, which is close to the theoretical and simulated radiation power (for additional details, see Supplementary Information Section S6). For comparison, the chip with the same structure but without the HMM has no obvious radiation peak and an output power of only 10–25 nW, which indicates that the detected THz wave is originated from CR other than Smith-Purcell radiation, hot-carrier-induced plasmon, or other effects (see more measurement results and discussions in Supplementary Information Section S7). Therefore, all the theoretical analysis and the controlled experiments indicate that on-chip free-electron-based CR in the THz region is experimentally realized, and moreover, the frequency can be tuned over a superwide range.

To further understand the on-chip CR in the THz region and the tunability properties of the chip, Fig. 2b, c sketches the wavevector matching between the evanescent field around the free-electron beam, CR modes in the HMM, and radiation coupled into free space. The black solid lines and colored solid lines represent the dispersion curve of the evanescent field surrounding a free-electron (namely, $\omega = u_0 \cdot |\mathbf{k_z}|$) and the projections of the $k_x$-$k_z$ hyperbolic curve in Fig. 1b onto the $\omega$-$k_z$ plane at different frequencies, respectively. In Fig. 2b, the black lines and colored lines intersect at the deep red, red, and light red points, which indicate the end points of the CR wavevector along the $z$-axis ($\mathbf{k_{CR-z}}$) in the HMM and have the same value. To extract the CR into free space, $\mathbf{k_z}$ needs to be matched by the wavevector introduced by the grating ($2\pi n/p$) to change it to the free space radiation region ($|\mathbf{k_z}| < \omega/c$), where only the $n = 1$ condition is considered due to the greatest intensity of the first-order diffraction. The three intersection

points reveal that a higher electron velocity (energy) results in a higher radiation frequency of the chip. This is verified by the measurement results in Fig. 2d, which show the measured radiation spectra with central frequencies of $f = 3.95$ THz ($E = 1.4$ keV, deep red line), 4.96 THz (2.0 keV, red line), and 5.61 THz (2.6 keV, light red line) when fixing $p = 5$ µm.

Figure 2c shows how the grating period affects the radiation frequency. The black line ($\omega = u_0 \cdot |\mathbf{k_z}|$) intersects with the red, violet, and yellow solid lines (projections of the $k_x$-$k_z$ hyperbolic curve onto the $\omega$-$k_z$ plane) at the corresponding points, respectively, which indicate the end points of the CR wavevector along the $z$-axis ($\mathbf{k_{CR-z}}$). The different $\mathbf{k_{CR-z}}$ values require different grating periods $p$ to extract the CR into free space, and a smaller $p$ corresponds to a higher radiation frequency for the same electron energy, which is verified by Fig. 2e. With a fixed $E = 2$ keV and a decreased grating period, the measured spectra from different chip units illustrate that the central frequencies of radiation coupled into free space are $f = 4.96$ THz, 6.55 THz, and 8.04 THz for $p = 5$ µm (red line), 4 µm (violet line) and 3 µm (yellow line), respectively. By the way, the measured linewidth of each peak in Fig. 2 is approximately 40 GHz (see Supplementary Information Section S8).

For the fixed $p = 2$ µm, the measured tunable range can reach over 5 THz (8.93–13.94 THz), which is one of the widest among the small-size THz sources based on free-electron radiation. Based on this, a chip array with different grating periods covering different frequency ranges, shown in Supplementary Fig. 9 of Supplementary Information Section S9, is fabricated. By merely switching the power supply on different units and adjusting the voltage, an on-chip integrated free-electron THz source with a tunable frequency range spanning 3.2–14 THz is achieved. Compared with other free-electron THz sources, the tunable range is comparable to that of the most advanced accelerator-based tunable THz sources[22], while the chip size is more than three orders of magnitude smaller. Moreover, compared with tunable photonic THz sources[30–35], the radiation chip has significant advantages of a superwide, continuous, and electrical tunable range, which can hardly be achieved in gas lasers[33], QCLs[30–32], or other photonic methods[34,35].

## Analysis

Furthermore, theoretical calculations (according to the relation $\omega/u_0 = 2\pi n/p$) and numerical simulations (using the particle-in-cell finite difference time domain method; see more details in Supplementary Information Section S10) are conducted for comparison with the measured results. Figure 3a shows the calculated (lines), simulated (crosses), and experimentally measured (hollow circles) radiation frequency of the chip with varying $E$ and $p$. When the electron energy increases from 1.4 to 2.6 keV, the theoretical tunable ranges of a single unit with fixed grating structure are about 1.3–4 THz, which are 3.69–5.02 THz ($p = 6$ µm, orange line), 4.43–6.03 THz ($p = 5$ µm, red line), 4.92–6.69 THz ($p = 4.5$ µm, brown line), 5.54–7.53 THz ($p = 4$ µm, light blue line), 6.33–8.61 THz ($p = 3.5$ µm, deep blue line), 7.38–10.04 THz ($p = 3$ µm, purple line), and 11.07–15.06 THz ($p = 2$ µm, gray line). As $p$ increases, the electrically tunable range shrinks, which obstructs further expansion of the tunable range to lower frequencies.

The simulated radiation frequency is depicted as crosses in Fig. 3a, which is highly consistent with the theoretical lines considering the small error caused by the slight divergence of the electron beam and mesh influence. Taking the chip with $p = 5$ µm as an example, Fig. 3b depicts the simulated spectrum of the CR inside the HMM (black line) and that coupled into free space under $E = 1.4$ keV (deep red line), 2.0 keV (red line), and 2.6 keV (light red line). Figure 3c depicts the simulated spectra of three different chips with $p = 5$ µm (red line), 4 µm (violet line), and 3 µm (yellow line) when the electron energy is fixed at $E = 2$ keV. The simulated spectra and central frequency of radiation coupled into free space shown in Fig. 3b, c show great consistency with the measured results in Fig. 2d and the theoretical results in Fig. 3a. The

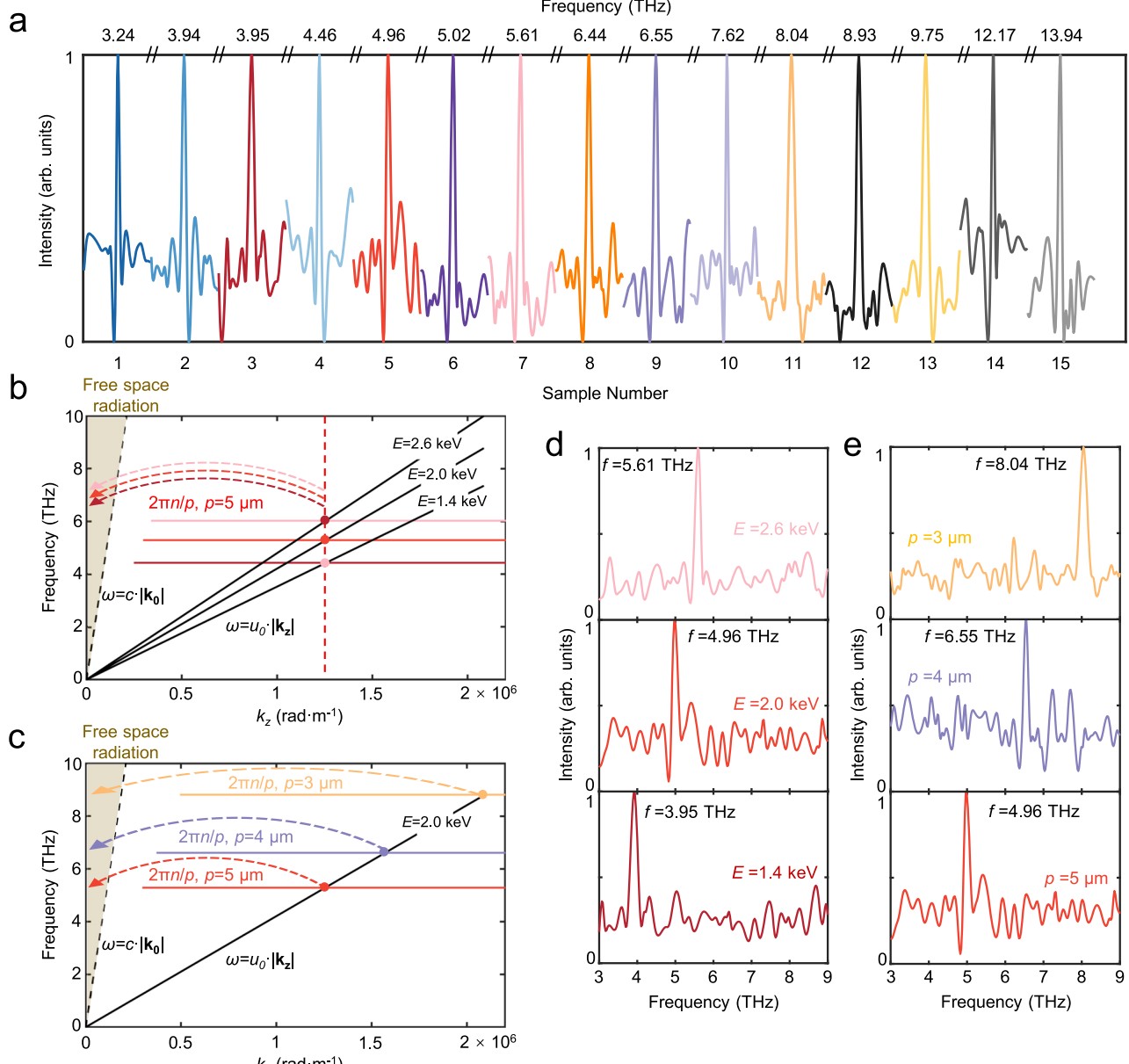

**Fig. 2 | Measurement results of the THz radiation frequency and spectrum of the chip. a** Measured spectral peaks spanning 3.2–14 THz for different grating periods under different electron energies. The central frequency and the sample numbers are shown at the top and bottom of the figure, respectively. The corresponding values of $E$ and $p$ for different samples are listed in Supplementary Information Section S5. **b** Wavevector matching for the CR in the HMM and radiation coupled into free space. The deep red, red, and light red solid lines represent the projections of the $k_x$-$k_z$ hyperbolic curve (the red hyperbolic curve indicated in Fig. 1b) onto the $\omega$-$k_z$ plane at different frequencies. The black lines are the dispersion lines of the evanescent fields surrounding free electrons (namely, $\omega = u_0 \cdot |\mathbf{k_z}|$) with different $u_0$ ($E$). The black lines and colored lines intersect at the deep red, red, and light red points, which indicate the end points of the $\mathbf{k_z}$ of the excited CR in the HMM and correspond to the same $\mathbf{k_z}$. To extract the CR into free space, $\mathbf{k_z}$ needs to be matched by the wavevector introduced by the grating ($2\pi n/p$)

to change it to the free space radiation region ($|\mathbf{k_z}| < \omega/c$). According to this figure, a higher electron velocity (energy) results in a higher radiation frequency of the chip. **c** The black line ($\omega = u_0 \cdot |\mathbf{k_z}|$) intersects with the red, violet, and yellow solid lines (projections of the $k_x$-$k_z$ hyperbolic curve onto the $\omega$-$k_z$ plane) at the red, violet, and yellow points, which indicate the end points of the $\mathbf{k_z}$ of the excited CR and correspond to different $\mathbf{k_z}$ values. To extract the CR into free space, $\mathbf{k_z}$ needs to be matched by the wavevector introduced by the gratings ($2\pi n/p$) of different grating periods $p$, and a smaller $p$ corresponds to a higher radiation frequency. **d** Measured radiation spectra at $E = 1.4$ keV (deep red line), 2.0 keV (red line), and 2.6 keV (light red line). The grating period of the chip is fixed at 5 μm. **e** Measured radiation spectra at $p = 5$ μm (red line), 4 μm (violet line), and 3 μm (yellow line). The electron energy is fixed at 2 keV. Each spectrum in this figure is normalized by its own maximum peak.

measured results show slight redshifts compared to the calculated line, which is mainly due to insufficient electron acceleration (see Supplementary Section S11). Based on the measured results, for an arrayed chip, four units are needed at least to cover the 3.2–14 THz range. Figure 3d depicts the simulated field contours (normalized $|\mathbf{E_z}|$ component) of the THz CR in the HMM. The top row of three contours

of different frequencies in Fig. 3d shows that when $E$ is fixed, there exists a wideband CR in the HMM, and the field distribution changes with frequency. Additionally, the bottom row of three contours of different frequencies in Fig. 3d shows that for a certain target frequency, the field contour will change with $E$, which means that the period of the grating used to extract the radiation should be

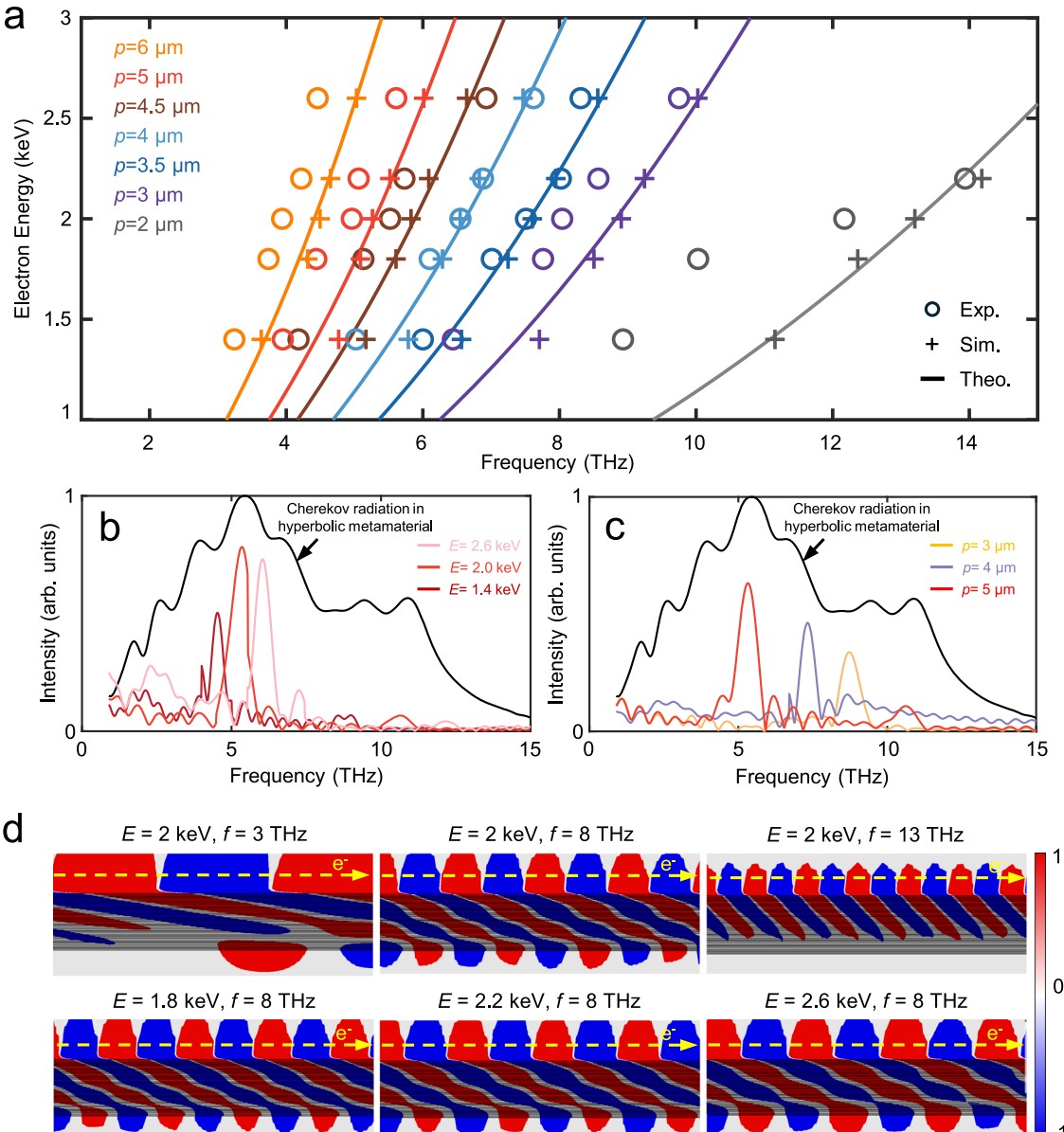

**Fig. 3 | Theoretical and simulation analysis of the generation of radiation with a selectable frequency. a** Frequency dependence on the grating period and electron energy. The solid lines, crosses, and hollow circles denote the theoretical, numerical simulation, and experimental results of the radiation frequency, respectively. The change in $p$ is represented by different colors. The measured tunable range could cover the frequency range of 3.2–14 THz. **b** Simulated spectra of the THz radiation when $p$ is fixed to 5 μm and $E$ = 1.4 keV (deep red line), 2.0 keV (red line), and 2.6 keV (light red line). The black line is the normalized spectrum of the broadband CR in the HMM, and the colored lines are spectra coupled into free space. **c** Simulated spectra of THz radiation when $E$ is fixed at 2 keV and $p$ = 5 μm (red line), 4 μm (violet line), and 3 μm (yellow line). The black line is the normalized spectrum of the wideband CR in the HMM, and the colored lines are the spectra coupled into free space. **d** Simulated CR field contour (normalized $|\mathbf{E_z}|$ components) in the HMM. The first row corresponds to $E$ = 2 keV, with $f$ = 3 THz, 8 THz, and 13 THz, which shows that a wideband CR field exists in the HMM. The second row corresponds to $f$ = 8 THz, with $E$ = 1.8 keV, 2.2 keV, and 2.6 keV. The CR field distribution changes with increasing electron velocity, which means that the period of the extraction grating needs to correspondingly change.

correspondingly adjusted. As a result of the combined effect of $E$ and $p$, frequency-selectable THz radiation can be extracted from the wideband CR in the HMM, as shown by the above experimental results.

## Discussion

In conclusion, we experimentally observe on-chip, free-electron-based CR in the THz region and realize an on-chip free-electron THz radiation source with superwide frequency tunability. By constructing a multilayer graphene/hBN HMM and integrating planar Mo cathodes and a grating on a chip, THz radiation with a maximum dynamically tunable

range of over 5 THz is excited by low-energy free electrons of 1.4–2.6 keV. A total frequency tunable range covering 3.2–14 THz is realized on a chip, and the central frequency can be continuously tuned by simply varying and switching the voltage between the electrodes. The tunable range is one of the widest among the free-electron THz sources and is comparable to large-scale accelerator-based FELs. The core area of the chip is only approximately 100×100 μm, which is more than three orders of magnitude smaller than that of even compact vacuum-tube free-electron THz sources. This work observes the on-chip free-electron-based CR in the THz region, provides a method to realize on-chip integrated THz sources with a superwide tunable

frequency range, and is promising for further extending free-electron radiation to a wider frequency range on a chip.

## Methods

### Sample fabrication

Quartz with high smoothness (surface roughness below 0.5 nm) and high THz transmittance is adopted as the substrate. After ultraviolet photolithography, a $SiO_2$ layer with a thickness of ~350 nm and a Mo layer with a thickness of ~350 nm are sequentially sputtered, followed by a lift-off process to form planar Mo electrodes. The $SiO_2$ gap layer prevents collision between the electron beam and the HMM, and the cathode with a zigzag shape at the front edge has a row of tips with a curvature radius of 2 μm for generating a belt-like free-electron beam. Then, a grating with a period of 2–6 μm and a slit width of 1 μm is fabricated on the substrate and between the electrodes by photolithography, sputtering of a Cr (thickness of 2 nm)/Au (thickness of 50 nm) layer, and etching with a focused ion beam.

The last step is to prepare a multilayer THz HMM and transfer it onto the chip. The graphene (thickness of 1–5 nm) and hBN (thickness of 40–50 nm) layers are obtained through mechanical exfoliation from highly oriented pyrolytic graphite and a bulk hBN crystal. Using the PDMS-supported transfer method[36], graphene and hBN layers are stacked in an alternating manner onto the grating to form a four- to six-layer multilayer HMM. The thickness of each layer is measured by atomic force microscopy. To prevent short circuiting and burning, the HMM cannot contact the Mo electrodes, and to improve the flatness of the HMM, thermal annealing (300 °C for 2 h in a vacuum environment) is conducted. After fabrication, the chip is preserved in a vacuum environment (pressure $<10^{-6}$ Pa) to avoid oxidation and contamination.

### Measurement setup

A custom-made THz measurement system is developed for the experiment. The radiation chip is fixed to a stage with the planar electrodes connected to an external circuit with probes and wires. The circuit part contains a DC voltage source with a maximum voltage of 5 kV and a DC ammeter to measure the electron current emitted by the planar free-electron emitter. The chip and stage are located in the middle of a hemispherical stainless steel vacuum chamber (pressure $<10^{-6}$ Pa). At the top of the chamber, there exists a quartz window (with a radius of 1 inch and a thickness of 5 mm) for radiation collection. The THz optical path contains two TPX lenses (radius of 1 inch), one single-sided gold-plated reflector (radius of 1 inch), and one chopper (effective aperture of 1 inch, chopping frequency of 15 Hz). Through these optical elements, THz radiation can be collected, focused, and modulated.

A THz Golay detector (GC-1P) is installed and calibrated at the end of the optical path to detect the radiation intensity. For the spectrum measurements, a THz scanning Fabry–Perot interferometer (TSFPI) is set in front of the Golay cell to interfere with the radiation and measure the spectrum through a Fourier transform. With a pair of HRSi lenses (radius of 1 inch), a minimum scanning step of 1.25 μm, and a maximum cavity length of 9.5 mm, the TSFPI can be used to measure frequencies from 0.1 to 15 THz. In the experiment, the chopper, Golay cell, and TSFPI are controlled by a computer with integrated software, and the spectrum can be measured almost in real time.

## Data availability

The data for Figs. 2a, d, e and 3a–c are provided in the source data file. All the processed data of this work are provided within the main text, Supplementary Materials, and the source data file. The raw data are available from the corresponding author upon request. Source data are provided with this paper.

## Code availability

The codes used for calculation are available from the corresponding author upon request.

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

## Acknowledgements

This work was supported by the National Key Research and Development Program of China (2024YFA1209202, F.L.). The authors would like to thank Dr. Hanqi Feng, Dr. Jinyu Li, Dr. Hongteng Lin, Dr. Zhexuan Wang, Dr. Jiahao Tian, Mr. Junjie Liu, and Mr. Yakun Huang for their help and advice on this study.

## Author contributions

T.L. and F.L. contributed equally to this work. F.L. proposed the idea of a THz source based on CR in HMM and directed T.L., Y.C., and X.X. to the research work. T.L. and F.L. performed the theoretical analysis. T.L., Y.C., and X.X. performed the numerical simulations. F.L. and T.L. designed the device and experiment. T.L., Y.C., and X.X. fabricated the samples and carried out the measurement. T.L., F.L., Y.C., X.X., K.C., X.F., W.Z., and Y.H. discussed the results. T.L., F.L., and Y.H. wrote the manuscript, and revisions were made by all. F.L. and Y.H. led the overall direction of the project.

## Competing interests

The authors declare no competing interests.
