## [Transparent Peer Review file · Nature Communications]

On-chip Cherenkov radiation tuning in 3.2-14 THz

Corresponding Author: Professor Fang Liu

Version 0:

Reviewer comments:

Reviewer #2

(Remarks to the Author)

I appreciate the authors' effort to revise the manuscript with new experimental and theoretical supports. The controlled experiment in Supplementary S7 supports the argument that the main THz radiation is from Cherenkov radiation, but not other types of radiation. The new information about the array device and the radiation power also make the manuscript's argument more solid. I can recommend the acceptance of this manuscript by Nature Communications after authors answer the following minor questions.

1. The shift of the focus of the manuscript on the innovation of THz CR in HMM instead of state-of-the-art frequency-tunable THz source is valid. However, there is still some overstatement in the main text, e.g., "the first observation of the on-chip CR in the THz frequency region" in line 52. There have been many works of on-chip THz Cherenkov radiation, like [T1]. Statement like "On-chip, free-electron-based CR" is more appropriate.

2. Metallic gratings, while providing momentum compensation for THz wave extraction, are also lossy at THz frequencies. Can other type of gratings, e.g. dielectric gratings, work for THz wave extraction?

[T1] Suizu, K. et al. Extremely frequency-widened terahertz wave generation using Cherenkov-type radiation. Optical Express (2009).

Version 1:

Reviewer comments:

Reviewer #2

(Remarks to the Author)

The authors have fully answered my questions in the second revision. They focus the work on on-chip, free-electron-based Cherenkov radiation, and justified the use of metallic gratings for THz wave extraction. I recommend its publication on Nature Communications.

Response letter to the reviewers

Reviewer 2

We would like to express our sincere gratitude to your recommendation for the acceptance of our manuscript. You raised 2 minor questions and we are pleased to provide further explanation and analysis on them to improve the manuscript.

Comment 1: *“The shift of the focus of the manuscript on the innovation of THz CR in HMM instead of state-of-the-art frequency-tunable THz source is valid. However, there is still some overstatement in the main text, e.g., “the first observation of the on-chip CR in the THz frequency region” in line 52. There have been many works of on-chip THz Cherenkov radiation, like [T1]. Statement like “On-chip, free-electron-based CR” is more appropriate.”*

We agree with your suggestion that the description “on-chip, free-electron-based CR” is indeed more rigorous and precise. Thus, the modifications have been made to the full text which are highlighted in the comparison version of new manuscript.

Comment 2: *“Metallic gratings, while providing momentum compensation for THz wave extraction, are also lossy at THz frequencies. Can other type of gratings, e.g. dielectric gratings, work for THz wave extraction?”*

We believe the metallic grating is important and could hardly be replaced by the dielectric grating. To realize large in-plane scattering, especially with a limited number of grating periods, high contrast grating plays an important role which can realize detectable extracted radiation. However, dielectric grating can be relatively treated as the low contrast grating which is hard to realize efficient radiation extraction. To validate this point, we conducted a simulation with gold-vacuum and hBN-vacuum grating (hBN is a kind of dielectric in THz frequency region), respectively, with the same set-up (grating period is 5 μ m, width of slot is 1 μ m, other set-up stays consistent with Supplementary Information S10). The results show that the gold-vacuum grating shows a 7.5 times enhancement of the electric field intensity of the extracted radiation.

Thanks to this valuable question, the corresponding description has been supplemented in the 5th paragraph:

“Additionally, gold is adopted as the material of the grating because it shows metallic properties in the THz region and thus the grating shows high contrast property which can achieve large in-plane scattering, especially with a limited number of grating periods”